# Network Pharmacology and Molecular Docking Analysis of Active Compounds in Tualang Honey against Atherosclerosis

**DOI:** 10.3390/foods12091779

**Published:** 2023-04-25

**Authors:** Ain Nabila Syahira Shamsol Azman, Jun Jie Tan, Muhammad Nazrul Hakim Abdullah, Hasnah Bahari, Vuanghao Lim, Yoke Keong Yong

**Affiliations:** 1Department of Human Anatomy, Faculty of Medicine and Health Sciences, Universiti Putra Malaysia, Serdang 43400, Selangor, Malaysia; 2Advanced Medical and Dental Institute, Universiti Sains Malaysia, Bertam, Kepala Batas 13200, Penang, Malaysia; 3Department of Biomedical Science, Faculty of Medicine and Health Sciences, Universiti Putra Malaysia, Serdang 43400, Selangor, Malaysia

**Keywords:** atherosclerosis, Tualang honey, network pharmacology, molecular docking, bee product

## Abstract

Atherosclerosis, a pathological condition marked by the accumulation of lipids and fibrous substances in the arterial walls, is a leading cause of heart failure and death. The present study aimed to utilize network pharmacology to assess the potential pharmacological effects of bioactive compounds in Tualang honey on atherosclerosis. This is significant as previous studies have indicated the cardioprotective effects of Tualang honey, yet a comprehensive evaluation using network pharmacology has yet to be conducted. The bioactive compounds in Tualang honey were screened and the potential gene targets for these compounds were predicted through Swiss Target Prediction and SuperPred databases. Atherosclerosis genes were retrieved from the OMIM, DisGeNet, and GeneCards databases. The interaction between these compounds and atherosclerosis genes was established through protein–protein interaction, gene ontology, and KEGG pathway analysis. The results of these analyses were then further confirmed through molecular docking studies using the AutoDock Tools software. The results revealed that 6 out of 103 compounds in Tualang honey met the screening criteria, with a total of 336 potential gene targets, 238 of which were shared with atherosclerosis. Further analysis showed that these active compounds had a good affinity with key targets and were associated with biological processes related to protein phosphorylation and inflammation as well as pathways related to lipid and atherosclerosis and other signaling pathways. In conclusion, the study provides insight into the potential pharmacological effects of Tualang honey bioactive compounds on atherosclerosis, supporting its use as a promising treatment for the disease.

## 1. Introduction

Cardiovascular disease (CVD) is a major health concern worldwide, with approximately 19.9 million deaths in 2020 [1]. It is projected that this trend will continue and CVD will remain the leading cause of death by 2030, affecting nearly 23.3 million people globally [2]. In Malaysia, the situation is particularly alarming, as the onset of cardiovascular disease occurs at a younger age compared with other neighboring countries and Western nations, according to the National Cardiovascular Disease-Acute Coronary Syndrome (NCVD-ACS) Registry [3].

Atherosclerosis is a prevalent underlying pathology of CVD, characterized by plaque formation in the arterial walls. The pathogenesis of atherosclerosis involves a complex interplay of multiple stages—endothelial cell dysfunction, lipoprotein deposition and oxidation, inflammatory factor effects, and fibrous cap development [4]. Various risk factors, such as oxidative stress, high cholesterol levels, hypertension, diabetes, and smoking, have been implicated in the development of atherosclerosis [5]. The initial event that triggers this process is the transport of low-density lipoproteins (LDL) across the endothelium and into the arterial wall, which leads to the formation of foam cells and atheroma plaques [6]. Maintaining a healthy diet and lifestyle is crucial in reducing the risk of atherosclerosis, but medication therapy is also commonly prescribed. Statins are currently the most effective medication for treating this disease, as they help to lower levels of atherogenic lipoproteins and decrease the risk of major cardiovascular incidents. However, a small percentage of patients are unresponsive to statin therapy, and long-term use of these drugs may cause adverse effects particularly muscle-related problems [7,8], which has prompted the exploration of alternative treatments, such as the use of natural products.

Honey, with its rich content of polyphenols and other phytochemicals, has been used for medicinal purposes for centuries. The beneficial effects of honey on cardiovascular health have been attributed to several mechanisms, including improving lipid profiles [9], reducing oxidation of LDL cholesterol [10], providing anti-oxidant [11] and anti-inflammatory effects [12], which may help to reduce inflammation in the cardiovascular system. Studies on Malaysian honey samples have shown that Tualang honey has the highest content of phenolic and flavonoid compounds, which possess antioxidant properties, making it of significant scientific importance in human health and medicine [13]. Tualang honey is produced by Asian rock bees (*Apis dorsata*) from the branches of the Tualang tree (*Kompassia excelsa*) commonly found in the northwestern region tropical rainforests of Peninsular Malaysia [14]. Previous research has indicated that Tualang honey possesses the capability of mitigating dysfunction in vascular endothelial cells [15], a crucial pathological event in the progression of coronary atherosclerosis. Tualang honey may also confer cardioprotective effects against oxidative stress by inhibiting lipid peroxidation, thereby providing a promising approach for mitigating oxidative damage in the cardiovascular system [16,17]. Despite these discoveries, the precise targets and mechanisms by which Tualang honey functions to prevent atherosclerosis are not yet well understood.

In recent years, network pharmacology, aided by advancements in bioinformatics and related databases, has become a widely utilized tool in drug research to uncover the interrelationships between drugs, their targets, pathways, and related diseases [18]. Molecular docking analysis can be utilized by researchers to verify the potential associations between active components and target genes that have been identified through network pharmacology analysis [19]. Molecular docking involves superimposing small ligands onto the structures of macromolecular targets to uncover potential molecular interactions that may occur at the binding site. This method can provide a more in-depth understanding of the specific interactions between active components and the genes associated with various diseases, thus contributing to the development of potential therapeutic interventions.

In this study, we aim to investigate Tualang honey bioactive compound target and signaling pathways in the context of preventing atherosclerosis through a network pharmacology approach. By utilizing various databases, the complex relationships between the disease, drugs, and targets are systematically predicted and analyzed. The potential mechanisms of key targets for Tualang honey in treating atherosclerosis are also examined, with molecular docking techniques used to verify the potential targets of Tualang honey against atherosclerosis. The complete research process is depicted in Figure 1.

## 2. Materials and Methods

### 2.1. Bioactive Compounds of Tualang Honey

Identification of bioactive compound of Tualang honey were scholarly searched through the literature repositories such as PubMed (https://pubmed.ncbi.nlm.nih.gov, accessed on 26 January 2023) [20], Springer (https://link.springer.com, accessed on 26 January 2023) [21], and Science Direct (https://www.sciencedirect.com, accessed on 26 January 2023) [22].

### 2.2. Screening of Bioactive Compounds in Tualang Honey

In order to thoroughly understand the potential therapeutic benefits of a natural product, it is necessary to examine its human absorption, distribution, metabolism, and excretion (ADME) properties. To evaluate these parameters, an in silico integrative ADME model was utilized with the use of Traditional Chinese Medicine Systems Pharmacology Database (TCMSP; https://old.tcmsp-e.com/tcmsp.php, accessed on 30 January 2023) [23]. The screening process involved the elimination of compounds without ADME information. The assessment of crucial parameters, including oral bioavailability (OB) and drug-likeness (DL), aimed to identify potential bioactive compounds present in Tualang honey. The final list of compounds was determined based on criteria established by the TCMSP database, which requires a compound to have an OB ≥ 30 and DL ≥ 0.18 [24] in order to be considered as a potential bioactive component of Tualang honey.

### 2.3. Prediction of Target Genes in Tualang Honey Bioactive Compounds

The Swiss Target Prediction database (http://www.swisstargetprediction.ch/, accessed on 1 February 2023) [25] and the SuperPred database (https://prediction.charite.de/index.php, accessed on 1 February 2023) [26] were utilized to identify the target genes of the bioactive compounds found in Tualang honey. To ensure the relevance to human biology, the analysis of target genes was restricted to those found in Homo sapiens, with duplicated entries being eliminated to generate a final list.

### 2.4. Prediction of Target Genes in Atherosclerosis

The identification of target genes associated with the condition of atherosclerosis was sourced from three well-established databases: Online Mendelian Inheritance in Man (OMIM; https://www.omim.org/, accessed on 3 February 2023) [27], Disease Gene Interaction (DisGeNet; https://www.disgenet.org/, accessed on 3 February 2023) [28], and GeneCards (http://www.genecards.org/, accessed on 3 February 2023) [29]. The inquiry was conducted using the term “atherosclerosis” as the primary keyword and any redundant genes were removed to ensure a comprehensive and non-duplicative list.

### 2.5. Construction of Venn Diagram

The potential target genes for both Tualang honey and atherosclerosis were analyzed through InteractiVenn (http://www.interactivenn.net/, accessed on 4 February 2023) [30], with the aim of determining the target genes of Tualang honey that are related to atherosclerosis. The output of the analysis was presented in the form of a Venn diagram, which displayed the overlapping potential target genes between the Tualang honey and atherosclerosis.

### 2.6. Compound-Target Network

The establishment of a compound-target network was constructed through the utilization of the Cytoscape software version 3.9.1 (https://cytoscape.org/, accessed on 7 February 2023) [31] in order to more effectively illustrate the interactions between Tualang honey bioactive compounds and atherosclerosis. By categorizing and organizing the identified active components of Tualang honey and its respective target information, the data were then integrated into Cytoscape, taking the common target genes of both bioactive compounds of Tualang honey and atherosclerosis as the network nodes and representing the interconnections between these nodes through connecting lines. Ultimately, this process led to the formation of a “Compound-Target-Disease” network.

### 2.7. Protein–Protein Interaction (PPI) Network Analysis

The intersection between the target genes of bioactive compounds in Tualang honey and atherosclerosis were analyzed through a protein–protein interaction network analysis, utilizing the STRING database version 11.5 (https://string-db.org/, accessed on 7 February 2023) [32]. The analysis was limited to proteins within the *Homo sapiens* species and only those with a confidence score of at least 0.900 were considered for network visualization. The resulting network was depicted using the Cytoscape platform, integrating data obtained from the STRING database. The significance of genes in the network was determined through the utilization of the CytoHubba plugin [33], employing three algorithms, namely, maximal clique centrality, maximum neighborhood component, and degree—to identify the top 10 hub genes based on the scores. The results of the three algorithms were then intersected to obtain the final set of hub genes, thereby presenting them as potential hub targets in the network.

### 2.8. Gene Ontology and Pathway Enrichment Analysis

A gene ontology (GO) and Kyoto Encyclopedia of Genes and Genomes (KEGG) pathway enrichment analysis was conducted to gain a deeper understanding of the biological mechanisms underlying the combination of Tualang honey bioactive compounds and its potential effects against atherosclerosis. The analysis was performed using the Database for Annotation, Visualization, and Integrated Discovery (DAVID), version 2021 (https://david.ncifcrf.gov/home.jsp, accessed on 9 February 2023) [34]. The software provides a thorough repository for gene functional analysis, providing extensive information on biological processes, cellular components, molecular functions, and signaling pathways [35,36,37]. The results of the analysis were presented, with the top 15 from the GO analysis and KEGG analysis being shown. The ***p***-values were calculated, and statistically significant results were indicated by a *p*-value of less than 0.05.

### 2.9. Molecular Docking

The molecular docking approach was employed to analyze the interactions between the bioactive compounds of Tualang honey and the hub genes identified in the study. This computational method is commonly utilized in drug discovery as it enables predictions of binding modes and affinities between receptors and ligands. The crystal structures of the six hub genes—SRC (PDB ID: 1O43), PIK3R1 (PDB ID: 315R), PIK3CA (PDB ID: 6PYS), EGFR (PDB ID: 8A27), PTPN11 (PDB ID: 3B40), and AKT1 (PDB ID: IUNQ) were obtained from the RCSB Protein Data Bank (https://www.rcsb.org/, accessed on 13 February 2023) [38]. The 3D structure of the Tualang honey compounds (catechin, ethyl oleate, fisetin, hesperitin, kaempferol, and luteolin) was retrieved from the PubChem database (https://pubchem.ncbi.nlm.nih.gov/, accessed on 14 February 2023) [39]. The compounds served as the ligands and the hub genes as the receptors. The receptors were prepared in Biovia Discovery Studio 2021 [40] by removing water molecules and existing ligand. The binding conformation between the ligand and receptor was predicted by AutoDockTools Version 4.2 (http://autodock.scripps.edu/, accessed on 16 February 2023) [41], and the binding energy, which is an outcome of molecular docking, was used to assess the potential of the ligand–receptor binding (typically, ≤−5 kcal/mol). Finally, the visualization of molecular interactions between the proteins and ligands was performed using Discovery Studio Visualizer 2021.

## 3. Results

### 3.1. Screening and Identification of Active Compounds in Tualang Honey

According to the literature search [13,42,43,44,45,46,47], a total of 103 candidate compounds have been identified in Tualang honey after duplicate were removed (Appendix A). However, only six compounds—catechin [13,44,46], ethyl oleate [44], fisetin [43], hesperetin [13,43] kaempferol [13,43,44,46], and luteolin [13,43,44], were selected as the potential bioactive compounds that met the necessary requirements for drug screening, and thus, they were chosen for further analyses (Table 1).

### 3.2. Target Gene Prediction of Tualang Honey and Atherosclerosis

A total of 336 target genes of the six bioactive compounds in Tualang honey were identified through the utilization of Swiss Target Prediction and SuperPred databases (Appendix A). Additionally, a comprehensive search of three databases, namely, Online Mendelian Inheritance in Man (OMIM), Disease Gene Interaction (DisGeNet), and GeneCards, yielded a total of 5192 target genes related to atherosclerosis (Appendix A).

### 3.3. Common Target of Tualang Honey and Atherosclerosis

The potential targets of Tualang honey against atherosclerosis were represented by the Venn diagram (Figure 2). The diagram showed 238 Tualang honey intersection target genes against atherosclerosis.

### 3.4. Compound-Target Network Constructions

A network representation of the interaction between Tualang honey bioactive compounds and its potential targets for treating atherosclerosis was created using the Cytoscape software and depicted as the compound-target network (Figure 3). This network revealed the relationship between the six bioactive compounds present in Tualang honey and their associated targets in relation to the management of atherosclerosis.

### 3.5. Protein–Protein Interaction (PPI) Network

A PPI network analysis was conducted to demonstrate the interaction between the Tualang honey bioactive compounds and their potential targets in relation to atherosclerosis (Figure 4). This analysis was performed using the STRING database and depicted using the Cytoscape software. The network consisted of 176 nodes and 501 edges were established after hiding disconnected nodes that represented proteins and protein–protein associations, respectively. The Network Analyzer plugin in Cytoscape was utilized to examine the degree values of nodes in a network, where the color of the circle varied in correspondence with the degree value. Our analysis revealed an average degree value of 5.693, with a group of 65 nodes exhibiting degree values surpassing the mean (Appendix A). Notably, these highly connected nodes may represent important targets for the therapeutic effects of Tualang honey. The CytoHubba plugin, incorporating the maximal clique centrality (MCC), maximum neighborhood component (MNC) and degree algorithms, were then used to further determine the top 10 hub genes in the network, which were deemed to be potential targets for Tualang honey in treating atherosclerosis. Six hub genes were identified by taking the intersection of these three algorithms—SRC, P1K3R1, P1K3CA, EGFR, PTPN11, and AKT1 (Figure 5).

### 3.6. Gene Ontology and Pathway Enrichment Analysis

In order to gain further insights into the biological processes, cellular components, molecular functions, and pathway underlying the therapeutic effects of Tualang honey bioactive compounds against atherosclerosis, Gene Ontology (GO) and Kyoto Encyclopedia of Genes and Genomes (KEGG) pathway enrichment analyses were conducted using the DAVID bioinformatics resources. The results of the GO analysis (Figure 6) revealed that the top biological processes associated with the bioactive compounds in Tualang honey targets were related to protein phosphorylation and inflammatory response. The cellular component analysis revealed that the targets were primarily involved in the plasma membrane, cytoplasm, and cytosol. In terms of molecular function, the top 15 significant enrichment terms included RNA polymerase III transcription factor activity, enzyme binding, and protein kinase activity. The KEGG pathway analysis (Figure 7) results showed that many targets of bioactive compounds in Tualang honey against atherosclerosis were enriched in signaling pathways, such as EGFR tyrosine kinase inhibitor pathways, neurotrophin signaling pathway, Ras signaling pathway, prolactin signaling pathway, and sphingolipid signaling pathway. On top of that, KEGG revealed that Tualang honey bioactive compound might play an anti-atherosclerotic role directly through the lipid and atherosclerosis pathway. The potential target and mechanism of Tualang honey bioactive compounds against atherosclerosis in the lipid and atherosclerosis pathway are shown in Figure 8.

### 3.7. Molecular Docking

An analysis was performed to explore the interactions between the selected six active compounds of Tualang honey (catechin, kaempferol, hesperetin, ethyl oleate, luteolin, and fisetin) and six potential target genes (SRC, PIK3R1, PIK3CA, EGFR, PTPN11, and AKT1) at a molecular level through molecular docking studies. The AutoDock Tools software was utilized to conduct the molecular docking studies and a total of 36 docking results were obtained, 32 of which had binding energies below −5.0 kcal/mol. The docking score is shown in Table 2. The lower the docking score, the greater the binding force between the compound and the protein. Luteolin, fisetin, hesperitin, catechin, and kaempferol play an important role in atherosclerotic treatment since they bind well to each target with binding energy lower than −5.0 kcal/mol. Results showed that most of the active ingredients had a strong binding affinity with the target gene PIK3CA. Luteolin, fisetin, hesperitin, catechin, and kaempferol bind to PIK3CA with docking scores of −10.67, −10.50, −10.33, −10.28, and −10.08, respectively. The results of this study imply that targeting PIK3CA could potentially play a crucial role in the management of atherosclerosis by Tualang honey bioactive compounds. The visualizations of the lowest binding energies between each target and the Tualang honey bioactive compounds were created using Biovia Discovery Studio (Figure 9).

## 4. Discussion

Atherosclerosis is a complex disease characterized by the accumulation of lipid-rich plaques in arterial walls, leading to the narrowing of the arteries which contributes to most cardiovascular diseases [48]. Statin therapy remains the most effective intervention for managing the progression of atherosclerosis, however, a subset of patients is non-responsive to the treatment and long-term use may result in adverse effects. As a result, there is a need to identify alternative treatments and understand their underlying mechanisms for the management of atherosclerosis.

Tualang honey has been shown to possess multiple pharmacological activities that may target different pathways and processes involved in atherosclerosis. Tualang honey has exhibited promising results in animal studies, including its ability to provide cardioprotective effects against oxidative stress [16] and to lower systolic blood pressure, triglycerides, and very-low-density lipoprotein (VLDL) [49]. Despite the promising outcomes of previous studies, a deeper understanding of the underlying mechanisms by which Tualang honey may act to prevent atherosclerosis is yet to be fully explored. Further research is needed to validate these findings and to fully comprehend the targets and mechanisms of action of Tualang honey in the management of atherosclerosis. With this aim, the present study utilized a network pharmacology and molecular docking approach to validate the positive impact of Tualang honey bioactive compounds on the treatment of atherosclerosis.

The results of this study show that 238 potential targets for the treatment of atherosclerosis with Tualang honey bioactive compounds have been identified through network pharmacology analysis. The six hub targets, SRC, PIK3R1, PIK3CA, EGFR, PTPN11, and AKT1, were also discovered in the network diagram as the potential targets. To determine the efficacy of bioactive compounds in Tualang honey as an active agent, it is crucial to consider the pharmacokinetic properties of its components. In this study, only compounds in Tualang honey with an oral bioavailability (OB) higher than 30% and drug-likeness (DL) index greater than 0.18 were considered to be pharmacokinetically active, as they are likely to be absorbed and distributed within the human body. Six compounds—catechin, ethyl oleate, fisetin, hesperetin, kaempferol, and luteolin, were selected from Tualang honey as bioactive compounds that met the requirements of the drug screening and may play a significant role in the treatment of atherosclerosis.

The results of the KEGG pathway enrichment analysis demonstrate that, in addition to the enrichment observed in the lipid and atherosclerosis pathways, the main compounds found in Tualang honey also target several other signaling pathways that are closely linked to the development of atherosclerosis. These pathways include the neurotrophin signaling pathway, Ras signaling pathway, EGFR-tyrosine kinase inhibition pathway, prolactin signaling pathway, and sphingolipid signaling pathway. Numerous studies have substantiated the role of the neurotrophin signaling pathway in regulating the response of vascular smooth muscle cells to injury, a key aspect in the pathogenesis of atherosclerosis [50]. Furthermore, the components of the renin-angiotensin system, namely, angiotensin II (Ang-II), ACE2, and angiotensin-1-7 (Ang-1-7), have distinct roles in atherosclerosis. Ang-II promotes atherosclerosis, while ACE2 and Ang-1-7 have anti-atherosclerotic effects. These components also impact endothelial function, oxidative stress, inflammation, cellular activity, and plaque stability [51]. Conversely, the inhibition of the epidermal growth factor receptor (EGFR) has been demonstrated to be effective in preventing the formation of atherosclerotic lesions, reducing inflammation, and suppressing the generation of reactive oxygen species and foam cell formation [52]. In vitro studies have also indicated that prolactin plays a role in the development of endothelial dysfunction in atherosclerosis through its ability to modulate the inflammatory response, stimulate vascular smooth muscle cell proliferation, and regulate mononuclear cell adhesion to the endothelium [53]. Numerous studies have revealed that sphingolipids play an important role in endothelial dysfunction and thus may promote the atherosclerotic processes [54].

A molecular docking verification was conducted on six key targets of the hub genes and six active ingredients of Tualang honey. The molecular docking results indicate that 5 out of 6 compounds in Tualang honey, which are luteolin, fisetin, hesperitin, catechin, and kaempferol, play an important role in atherosclerotic treatment since these compounds could bind stably to the active pocket of the key target with binding energy lower than −5.0 kcal/mol. These compounds are all flavonoids which have received considerable attention for their potential to positively impact cardiovascular health. This is attributed to their ability in the inhibition of low-density lipoprotein oxidation, enhancement of endothelial-dependent vasodilation, reduction of adhesion molecule and inflammatory marker levels, protection of nitric oxide and endothelial function during oxidative stress, and prevention of platelet aggregation [55]. Some studies have shown that treatment with catechin could attenuate the atherosclerosis lesion development [56,57]. Similarly, fisetin reduced the buildup of lipids in the plaque that causes atherosclerosis by decreasing the activity of proprotein convertase subtilisin/kexin type 9 (PCSK9) and lectin-like oxidized low-density lipoprotein receptor-1 (LOX-1). This helped to decrease the amount of oxidized LDL taken up by macrophages, ultimately reducing the formation of foam cells [58]. Hesperetin has also been demonstrated to inhibit the formation of foam cells and promote cholesterol removal [59]. Additionally, kaempferol has been shown to have anti-inflammatory properties, exerting its effects on cardiovascular health by modulating the gene and protein expression of inflammatory molecules [60]. The flavonoid, luteolin, has been demonstrated to have the ability to reduce the proliferation and migration of vascular smooth muscle cells, which in turn suppresses platelet function. This is achieved through its inhibition of angiogenesis, induced by vascular endothelial growth factor, through targeting the activity of the phosphatidylinositol 3-kinase (PI3K) signaling pathway [61].

The results of the molecular docking studies conducted in this study indicated that a majority of the bioactive components of Tualang honey had a robust binding affinity for PIK3CA. This was evidenced by the fact that luteolin, fisetin, hesperitin, catechin, and kaempferol bound to PIK3CA with binding energy values of −10.67, −10.50, −10.33, −10.28, and −10.08, respectively. The stability of the ligand–receptor binding conformation and the potential for interaction increase as the binding energy decreases [62]. These results suggest that PIK3CA might be a primary target for Tualang honey in the treatment of atherosclerosis. PIK3CA is a subset of an enzyme known as phosphatidylinositol 3-kinase (PI3K) and its activation plays a crucial role in the development of atherosclerosis through the abnormal proliferation and migration of smooth muscle cells leading to thickening of the arterial intima [63]. Prior research has revealed that inhibiting PI3K activity could be a promising strategy for the treatment of atherosclerosis, as demonstrated by the reduction of plaque size in the ApoE−/− model upon PI3K gene deletion [64]. Above all, according to our analysis, compounds of Tualang honey may targets the activity of the PI3K specifically its subset PIK3CA in the treatment of atherosclerosis as this target gene displayed a robust binding energy with all the compounds and was one of the central hub gene in the protein–protein interaction network. However, the compound of Tualang honey may also regulate other targets to effectively against atherosclerosis because they have an important position in the PPI network and displayed stable binding to the active pocket of the key protein.

Network pharmacology and the molecular docking approach were utilized in this study to investigate the potential active components, possible targets, and important biological pathways involved in the treatment of atherosclerosis by the polyphenolic compounds in Tualang honey. The results offer an initial theoretical foundation for further experimental investigation. However, it is worth mentioning that the pharmacological mechanism of Tualang honey bioactive compounds in the treatment of atherosclerosis is based on computational technologies and still requires further validation through pharmacological and clinical studies.

## 5. Conclusions

In summary, this study utilized a network pharmacology approach to investigate the potential mechanism of Tualang honey bioactive compounds in the treatment of atherosclerosis. Our results indicated that compounds in Tualang honey such as luteolin, fisetin, hesperitin, catechin, and kaempferol may have a crucial impact on atherosclerosis through their effects on key biological targets, including PIK3CA, SRC, P1K3R1, EGFR, PTPN11, and AKT1. Our molecular docking analysis also suggested that these active components of Tualang honey could interact effectively with these targets. While these findings provide a valuable basis for further investigation, it is important to note that further pharmacological and clinical research is required to validate our conclusions.

## Figures and Tables

**Figure 1 foods-12-01779-f001:**
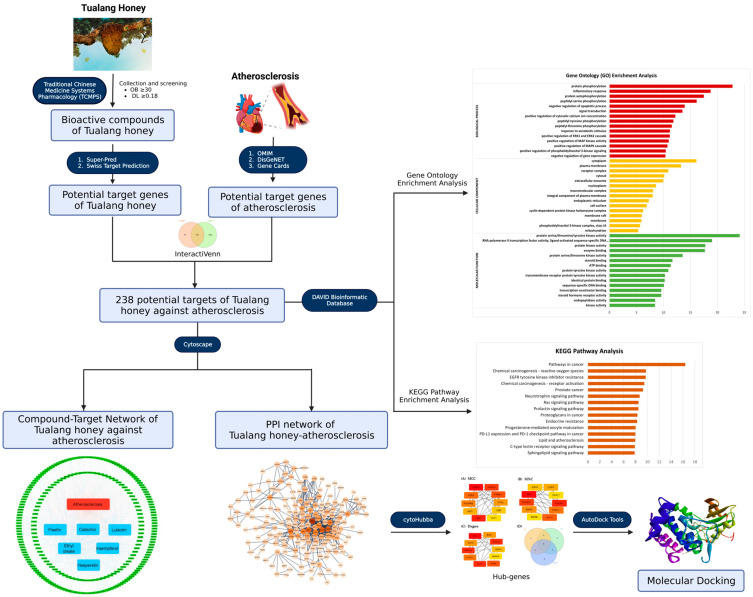
Network pharmacology and molecular docking workflow of bioactive compounds in Tualang honey for the treatment of atherosclerosis.

**Figure 2 foods-12-01779-f002:**
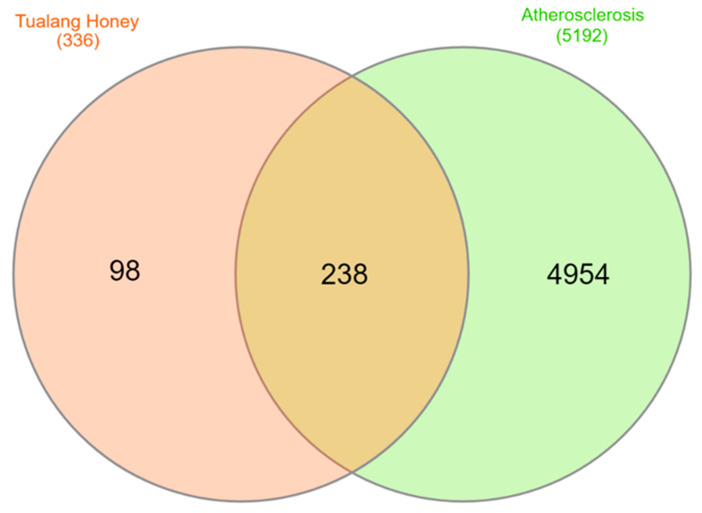
The potential targets of between six bioactive compounds of Tualang honey against atherosclerosis via Venn diagram. There are 238 target genes in Tualang honey bioactive compounds and atherosclerosis intersection (yellow).

**Figure 3 foods-12-01779-f003:**
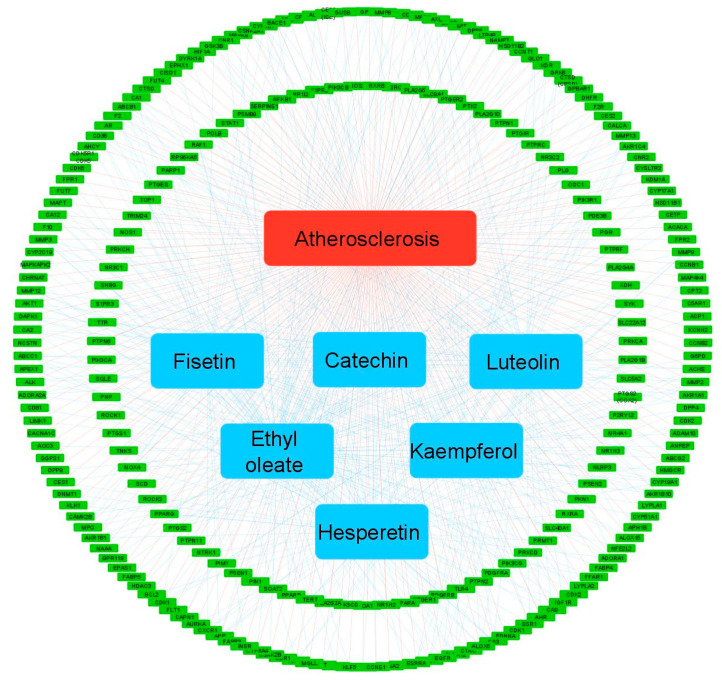
The compound-target network of Tualang honey bioactive compounds against atherosclerosis by using Cytoscape. The green nodes represent the interacting target genes between the compound (blue node) and target (red node).

**Figure 4 foods-12-01779-f004:**
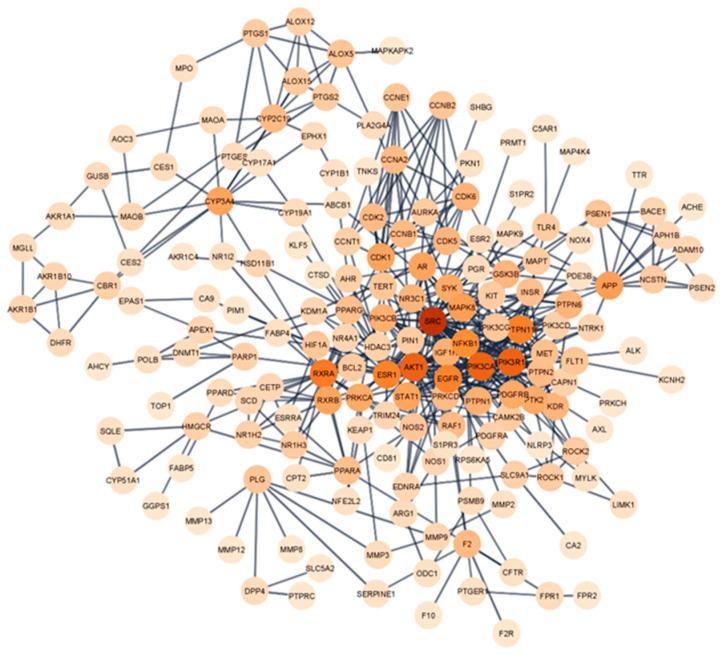
Protein–protein interaction networks are constructed using the STRING database. The degree value of nodes was visually represented in the network diagram using color gradients, with darker colors indicating higher values. The darker color of the node indicates that the target could play a role in the treatment of atherosclerosis by Tualang honey bioactive compounds.

**Figure 5 foods-12-01779-f005:**
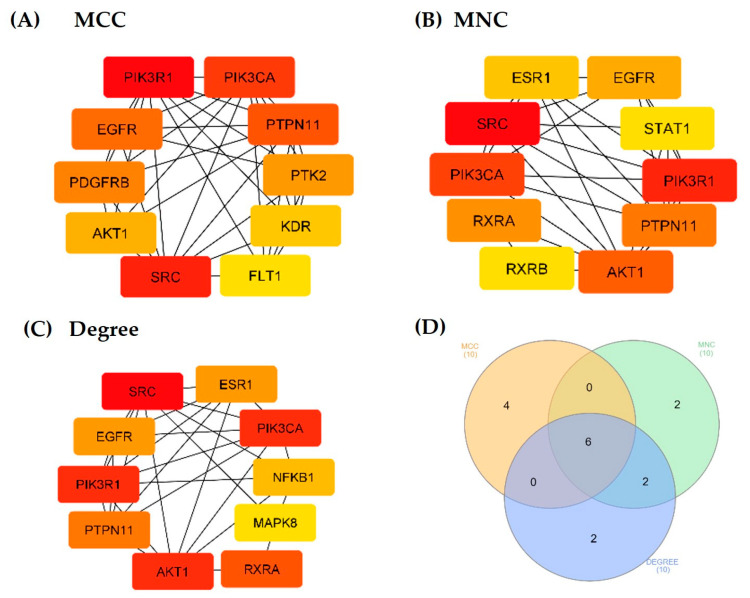
The top ten hub gene networks of Tualang honey bioactive compounds against atherosclerosis by incorporating three algorithms—(**A**) maximal clique centrality (MCC), (**B**) maximum neighborhood component (MNC), (**C**) degree of nodes, and (**D**) The Venn diagram illustrates six hub genes screened by the intersections of the three algorithms. The genes with the highest values are considered the most important hub genes and are depicted with a dark red color. Conversely, genes with lower values were considered less significant and are depicted with a light yellow color, indicating the ranking position of each gene in the network.

**Figure 6 foods-12-01779-f006:**
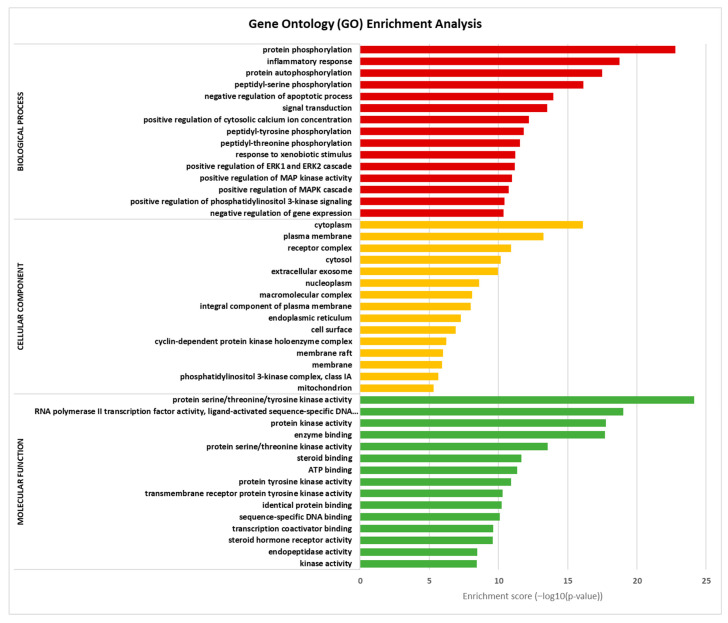
Gene ontology (GO) enrichment analysis. The bar chart represents the most significantly enriched GO terms (−log10 (*p*-value)) in biological processes, cellular components, and molecular function, comprising the top 15 terms related to the target genes.

**Figure 7 foods-12-01779-f007:**
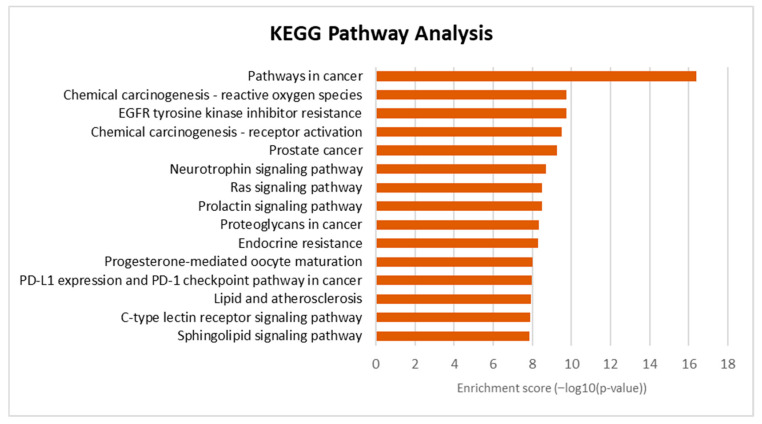
KEGG pathway enrichment analysis. The bar chart visualizes the top 15 enriched KEGG pathways of Tualang honey against atherosclerosis.

**Figure 8 foods-12-01779-f008:**
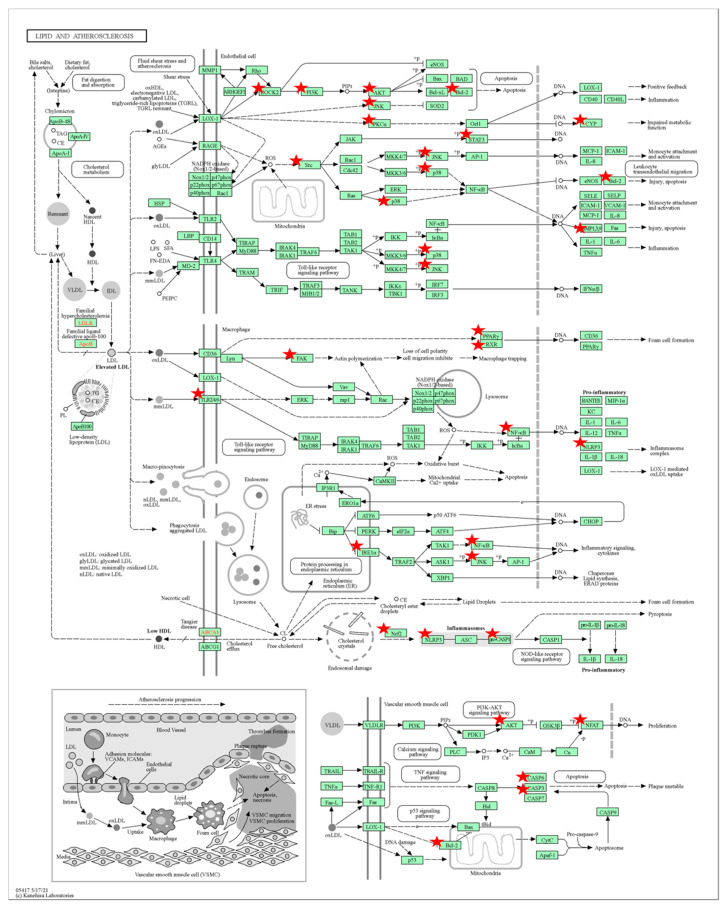
Potential targets and mechanism of bioactive compounds in Tualang honey against atherosclerosis. Red stars represent targeted genes involved in the lipid and atherosclerosis pathway.

**Figure 9 foods-12-01779-f009:**
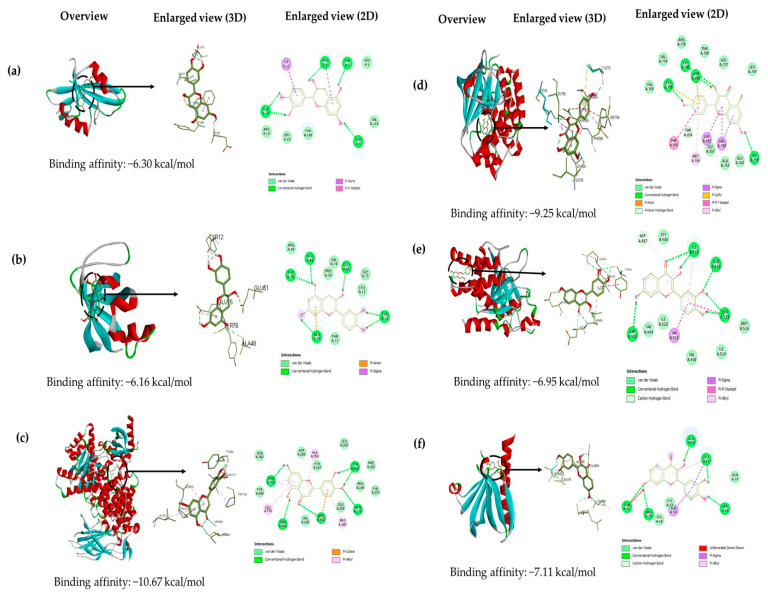
Molecular docking results of the lowest binding energy in each target with the bioactive compound of Tualang honey. (**a**) SRC-catechin, (**b**) PIK3R1-catechin, (**c**) PIK3CA-luteolin, (**d**) EGFR-kaempferol, (**e**) PTPN11-fisetin, and (**f**) AKT1-fisetin.

**Table 1 foods-12-01779-t001:** The pharmacokinetic properties of the bioactive compounds of Tualang honey based on Traditional Chinese Medicine Systems Pharmacology (TCMSP) database.

Compound	Oral Bioavailability OB (≥30%)	Drug-Likeness DL (≥0.18)
Catechin	54.83	0.24
Ethyl oleate	32.4	0.19
Fisetin	52.6	0.24
Hesperetin	70.31	0.27
Kaempferol	41.88	0.24
Luteolin	36.16	0.25

**Table 2 foods-12-01779-t002:** Binding energy between active compounds and six core targets of Tualang honey.

Compound	Binding Energy (kcal/mol)					
	SRC	PIK3R1	PIK3CA	EGFR	PTPN11	AKT1
Catechin	−6.30	−6.16	−10.28	−8.68	−6.78	−6.43
Ethyl oleate	−3.57	−3.45	−7.59	−6.63	−4.59	−4.58
Fisetin	−5.88	−5.71	−10.50	−9.19	−6.95	−7.11
Hesperetin	−5.84	−5.85	−10.33	−7.97	−6.55	−6.41
Kaempferol	−6.05	−5.94	−10.08	−9.25	−6.93	−6.70
Luteolin	−6.08	−5.99	−10.67	−8.01	−6.59	−6.98

## Data Availability

Data is contained within the article or Appendix A.

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
