# Peer review of "Network Pharmacology and Molecular Docking Analysis of Active Compounds in Tualang Honey against Atherosclerosis"

_foods, 2023, doi:10.3390/foods12091779_

Round 1

Reviewer 1 Report

The authors utilized a network pharmacology and molecular docking method to predict and validate the potential interactions between atherosclerosis (disease) and Tualang honey (drug), and obtained expected results. It was a meaningful work. However, in my opinion, it was not a complete job yet. In orde to ensure the scientific validity of the results, further animal experiments or clinical trials are required.

Author Response

We agree that network pharmacology is a quick and efficient method for predicting multiple drug targets in complex diseases, and it is still necessary to verify the predicted targets using in vitro or in vivo experiments. However, the aim of this current manuscript is to prove the concept before proceeding to the in vitro or animal work. Proving the concept before in vitro work is important because it helps ensure that the research is based on a solid foundation and has the potential to yield meaningful results. Without proving the concept, in vitro work may be misdirected, leading to wasted resources and potentially misleading results. In vitro work can also be time-consuming and expensive, so it is important to ensure that it is justified by a well-supported concept. Proving the concept can also help researchers identify potential pitfalls or limitations in their approach, allowing them to refine their methods and ensure that their in vitro work is more likely to be successful.

Reviewer 2 Report

1. Kindly elobarte the introduction

cite the current year references

like

2. Provide the better quality of figure 8 and 9

Reviewer 3 Report

foods-2298264-peer-review-v1

Although the text is very well written and the analyzes presented are of high relevance in view of the technology used, there are crucial errors that must be corrected, since there is a total exaggeration and distortion in the reading of the results. Therefore, I ask authors to pay special attention to the following points:

1-The authors did a sophisticated computational evaluation of 7 selected compounds found previously by other authors and available in the published literature.  I agree with the selected compounds that met the necessary requirements for drug screening, except for dioctyl phthalate which, in my point of view, does not belong to honey itself. This molecule could be a general contaminant (probably from pesticides and plastics) and should be taken from the list of results, in this manuscript.  The results at the end corroborate that only the five polyphenols luteolin, fisetin, hesperetin, catechin, and kaempferol play an important role in anti-atherosclerotic bioactivity, due to their bind to each target (binding energy lower than −5.0 kcal/mol)

2-The results obtained by the authors are only correlated to the five bioactive compounds present in Tualang honey, and not Tualang Honey itself. The amounts of such compounds will be in low concentration in the product, once honey, in general, is made of water, and free sugars, added with minor bioactive molecules (minerals, phenolic acids, and flavonoids, as is the case in this research). The results should be redone without the compound dioctyl phthalate.

3-For the Ethyl oleate, in Figure 9 no data appear… This compound, as a derivative of the fatty acid oleic, has a connection to thrombin and coagulation factor X, and its bioactivity thus should be evaluated carefully. Nevertheless, the low amounts in this honey maybe don´t have an impact, as maybe happen for the flavonoids in this study, due to this so minor concentration.

4-The extrapolation, from the network representation of the interaction between these compounds and “Tualang honey” and its potential targets for treating atherosclerosis, is not linear.  As pointed out above these structures are minorities in this product. I suggest that all over the text, the expression “Tualang honey” should be always changed to “Tualang honey bioactive compounds”. The potential targets for atherosclerosis are only due to the structure of these flavonoids, nothing more. “Tualang honey” is only the vehicle where they are transported.

For instance, in Figure 8. Potential targets and mechanism of Tualang honey against atherosclerosis. Red stars represent targeted genes involved in the lipid and atherosclerosis pathway.

Should be

Figure 8. Potential targets and mechanism of Tualang honey biocompounds (ethyl oleate, catechin, fisetin, hesperetin, kaempferol, and luteolin), against atherosclerosis. Red stars represent targeted genes involved in the lipid and atherosclerosis pathways.

5-In Discussion please change also,

“Tualang honey”

To

“Tualang honey polyphenolic biocompounds”

Is very clear that in line 306, where the authors say

“Tualang honey has been identified through network pharmacology analysis.”

Should be

“ Tualang honey polyphenolic biocompounds have been identified through network pharmacology analysis.”

6-Recommendation:

Due to the high level of skills in computational evaluation, the authors should include the relative percentage of the biocompounds in the main matrix (Tualang Honey) and then evaluate the amount of honey that should be taken to achieve such bioactivity. Only by doing that, they are able to establish a correct relationship between the bioactivity of each compound and the full matrix Tualang honey.

If this will be not possible in this manuscript, please do a sentence explaining this gap, and highlight the need for that, for a more accurate Reading of the data provided by all these tools.

Round 2

Reviewer 1 Report

A persuasive response has been madeand the revision is recommended to be accepted now.

Reviewer 3 Report

The determination of the relative percentage of bioactive compounds in Tualang Honey and the amount of honey needed to achieve their bioactivity would depend on multiple factors, including honey composition, processing, storage conditions, and stability of bioactive compounds. Obtaining precise information on the concentration of bioactive compounds in honey and their corresponding bioactivity would require further experimental analysis, including quantification using validated methods, in vitro assays, and in vivo studies.